# Therapeutic potential of hookworm proteins in promoting regulatory immune responses to modulate *Trypanosoma cruzi* induced liver inflammation and oxidative stress

**Maria Jose Villar[1,2], Cristina Poveda[1,2,3], Bin Zhan[1,2,3], Maya Gonsoulin[1], Kathryn M Jones[1,2,3]/+**

[1]Baylor College of Medicine, National School of Tropical Medicine, Department of Pediatrics, Houston, TX, United States
[2]Texas Children's Hospital Center for Vaccine Development, Houston, TX, United States
[3]Baylor College of Medicine, Department of Molecular Virology and Microbiology, Houston, TX, United States

**BACKGROUND** Chronic *Trypanosoma cruzi* infection causes significant liver pathology, and current antiparasitic treatments often worsen hepatic damage. Hookworm-derived proteins have shown immunomodulatory effects in inflammatory diseases, including *T. cruzi*-induced myocarditis.

**OBJECTIVE** This study evaluates recombinant hookworm proteins AIP-1 and AIP-2 for treating liver inflammation in a murine model of chronic Chagas disease (CD).

**METHODS** Female BALB/c mice infected with *T. cruzi* were treated with AIP-1 or AIP-2 (1 mg/kg) for seven days. Controls were untreated or received aspirin (25 mg/kg) for 14 days. Liver tissues were analyzed for parasite burden (quantitative polymerase chain reaction - qPCR), histopathology (H&E, Picrosirius Red), and cytokines (multiplex assay). Splenocytes were assessed by flow cytometry, and serum was tested for liver enzyme levels.

**FINDINGS** AIP-1 and AIP-2 increased hepatic interferon gamma (IFN-γ) and interleukin 10 (IL-10), decreased *Nfk-B* and *Stat*-1, and elevated *Arg*1 and *Nos*2 expression. AIP-1 uniquely upregulated *Mmp*9 and *Btg*2. Increased splenic CD11b⁺CD11c⁺ and CD11b⁺Ly6G^lo Ly6C⁺ cells were observed. Despite increased immune cell infiltration, parasite load and fibrosis remained unchanged, and liver enzyme levels were stable.

**MAIN CONCLUSION** AIP-1 and AIP-2 reduce hepatic inflammation and promote a balanced $T_H1/T_H2$ response, likely mediated by regulatory dendritic and myeloid-derived suppressor cells, supporting their potential as immunotherapeutic for *T. cruzi*-induced liver pathology.

Key words: immunomodulation - hookworm - *Trypanosoma cruzi* - liver

Chagas disease (CD), caused by the protozoan parasite *Trypanosoma cruzi*, typically spreads through contact with feces from infected triatomine vectors, commonly referred as 'kissing' bugs. This neglected tropical disease is primarily prevalent in rural areas of the Southern United States, Mexico, Central, and South America, affecting approximately 6-7 million people globally.[1] Rapid urbanization and migration from rural areas contribute to its spread beyond Latin America, resulting in cases emerging in regions such as Europe, Australia, and Japan.[2] Recognizing the signs and symptoms of this infection is crucial, particularly since the acute stage often goes untreated, potentially leading to complications in the heart, digestive tract, brain, and other organs.[3]

Acute CD typically presents with asymptomatic or mild, non-specific symptoms after an incubation period of one to two weeks.[4] As a result, it often progresses to the chronic stage, leading to severe complications due to tissue inflammation and degradation.[5] *T. cruzi* infections evade innate immunity, in part by resisting phagocytosis. This process modulates nitric oxide (NO) levels and activates macrophages and cytokines.[6] Such macrophage activation triggers a cascade of cytokines, including interleukin-2 (IL-2) and interferon-γ (IFN-γ), while regulatory cytokines such as IL-10 and IL-4 help mitigate potential harmful effects of an overactive immune response.[6] In experimentally infected animals, myeloid derived suppressor cells (MDSC) accumulate in the heart and spleen, regulating inflammatory T cell proliferation and promoting regulatory T cells. This mechanism helps control tissue damage while facilitating parasite persistence.[7,8]

Financial support: This work was funded by a Pilot Award from the Baylor College of Medicine Cardiovascular Research Institute and NIH R01AI168038 grant (KMJ). This project was supported by the Human Tissue Acquisition and Pathology core with funding from the Comprehensive Cancer Center grant (P30 CA125123), by the Cytometry and Cell Sorting Core at Baylor College of Medicine with funding from the CPRIT Core Facility Support Award (CPRIT-RP180672), the NIH (CA125123 and RR024574) and the assistance of Joel M Sederstrom, and the Center for Comparative Medicine Research Services Laboratory at Baylor College of Medicine.
+ Corresponding author: kathrynj@bcm.edu | ⓘ https://orcid.org/0000-0001-8745-1987

**Handling editor:** Adeilton Alves Brandão | ⓘ https://orcid.org/0000-0001-5877-607X

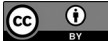

The only approved drugs for CD are benznidazole (BNZ) and nifurtimox, both of which have demonstrated a trypanocidal effect on the parasite. However, these antiparasitic treatments only reduce parasite load and may lead to toxic side effects, especially in chronic cases, when administered over an extended period.[9] BNZ is extensively metabolized in the liver by the cytochrome P450 enzyme complex, often resulting in Grade 1 or Grade 2 toxicity during treatment. It is also associated with elevated serum enzymes, such as AST and ALT.[10,11] Infection with *T. cruzi* can lead to severe cardiological and gastrointestinal complications and, in the presence of underlying conditions, infection can aggravate liver inflammation and accelerate hepatic injury.[12] Importantly, both *T. cruzi* infection and BNZ have been shown to stimulate oxidative stress in the liver, leading to pathologic remodeling.[13]

The liver plays a pivotal role in inflammatory responses, notably through the acute-phase response and oxidative stress mechanisms. When inflammation occurs, the liver produces acute-phase proteins (APPs), which indicate systemic inflammation. This process is mainly controlled by IL-6, a cytokine released by hepatic macrophages and other cells. IL-6 prompts hepatocytes to generate APPs such as C-reactive protein and fibrinogen.[14] The balance between positive APPs, which increase during inflammation, and negative APPs, which decrease, is essential for evaluating the body's inflammatory response. Due to its role in metabolizing various substances, the liver is particularly vulnerable to reactive oxygen species (ROS), increasing the risk of oxidative stress.

The persisting ROS release from a chronic *T. cruzi* infection leads to mitochondrial dysfunction, inflammation, and heart tissue damage.[15] Additionally, *T. cruzi* infection induced oxidative stress in the livers of infected mice.[16] Therefore, this stress may contribute to liver damage and further inflammation.[14,17] Drug-induced liver injury (DILI) arises when disruptions in drug metabolism trigger liver damage, often due to oxidative stress, inflammation, apoptosis, necrosis, or mitochondrial membrane breakdown, ultimately compromising organ function.[18] As an alternative approach to reduce inflammation caused by infectious pathogens or autoimmune diseases, helminth-derived proteins have demonstrated significant anti-inflammatory property. Previous studies have shown that canine hookworm *Ancylostoma caninum*-secreted AIP-1 and AIP-2 conferred therapeutic effect on colitis [19] and cardiac inflammation caused by chronic infection of *T. cruzi*.[20] In this study, we propose a novel treatment strategy utilizing these two hookworm-derived recombinant proteins (AIP-1 and AIP-2) provided immunomodulatory and antioxidant properties to treat *T. cruzi* infection caused liver inflammation and damage.[20]

### MATERIALS AND METHODS

*Ethics statement* - Animal experiments were performed in full compliance with the Public Health Service Policy and the National Institutes of Health Guide for the Care and Use of Laboratory Animals, 8th edition, under a protocol approved by Baylor College of Medicine's Institutional Animal Care and Use Committee, assurance number D16-00475.[21]

*Expression and purification of recombinant protein* - The recombinant proteins, AIP-1 and AIP-2, were expressed and purified as previously described.[20] Briefly, DNDs encoding for Ac-TMP-1 (AIP-1) and Ac-TMP-2 (AIP-2) were cloned into *Pichia pastoris* expression vector pPICZαA. The recombinant plasmid DNAs including AIP-1 and AIP-2 were transformed into *P. pastoris* X-33, the recombinant proteins (AIP-1 and AIP-2) were expressed under induction of 1% methanol and purified with immobilized metal affinity chromatography for AIP-1 (with His-tag at C-terminus) and ion exchange (HiTrap QXL) for AIP-2.

*Mice and parasites* - Male ICR-SCID (ICRSC-M) and female BALB/c (BALB/cAnNTac), five-six weeks of age from Taconic (Taconic Biosciences, Inc) were allowed to acclimate for approximately one week prior to initiating studies. Mice were housed four per group in small microisolator cages and provided ad libitum food and water, under a 12 h light/ dark cycle. To expand blood form trypomastigotes (bft) for experimental infection, male ICR-SCID mice were infected with 5000 bft of the bioluminescent *T. cruzi* H1 strain (*T. cruzi* H1 K68) generated in our laboratory[20,22] by intraperitoneal (IP) injection. Approximately 28 days after infection, ICR-SCID mice were humanely euthanized and bft were collected, washed with sterile medical grade saline, and used to infect BALB/c mice.

*Experimental infection and treatments* - Female BALB/c mice were infected with 5000 *T. cruzi* H1 K-68 bft by IP injection. At 70 days post-infection (dpi), animals were randomly assigned to treatment groups as outlined in Table.

As illustrated in Fig. 1, mice received daily IP injections of either AIP-1 or AIP-2 protein (1 mg/kg) for seven consecutive days. Control groups included uninfected mice, infected untreated mice, and infected mice administered aspirin (25 mg/kg) via drinking water *ad libitum* for 14 consecutive days. Aspirin served as a positive control, administered according to a published study,[23] to provide a benchmark for anti-inflammatory activity in *T. cruzi*-infected mice. At 84 dpi, liver, spleen, and serum samples were collected for comprehensive immunological analysis. These assessments included quantification of parasite burden, evaluation of liver histopathology, cytokine profiling via Luminex multiplex assay, relative gene expression analysis, splenic immune cell profiling by flow cytometry, and examination of serum for clinical abnormalities.

*Liver collection* - The liver was harvested using aseptic techniques in cold 1X phosphate-buffered saline (PBS) within a biosafety cabinet. The largest liver lobe was then divided into four sections for DNA, RNA, and protein analyses, snap-frozen, and stored at -80ºC until further use. The fourth liver section was designated for histopathological analysis, wherein it was stained with Hematoxylin and Eosin (H&E) and Picrosirius Red to assess inflammation and fibrosis, respectively.

TABLE

Study groups

| Group | Infection | Treatment | Duration | Route of administration |
|---|---|---|---|---|
| #1 | None | None | N/A | N/A |
| #2 | *Trypanosoma cruzi* H1 K68 | None | N/A | N/A |
| #3 | *Trypanosoma cruzi* H1 K68 | 1 mg/kg AIP-1 | Once daily for seven days | Intraperitoneal injection |
| #4 | *Trypanosoma cruzi* H1 K68 | 1 mg/kg AIP-2 | Once daily for seven days | Intraperitoneal injection |
| #5 | *Trypanosoma cruzi* H1 K68 | 25 mg/kg Aspirin | 14 days | *Ad libitum* drinking water |

N/A: not applicable.

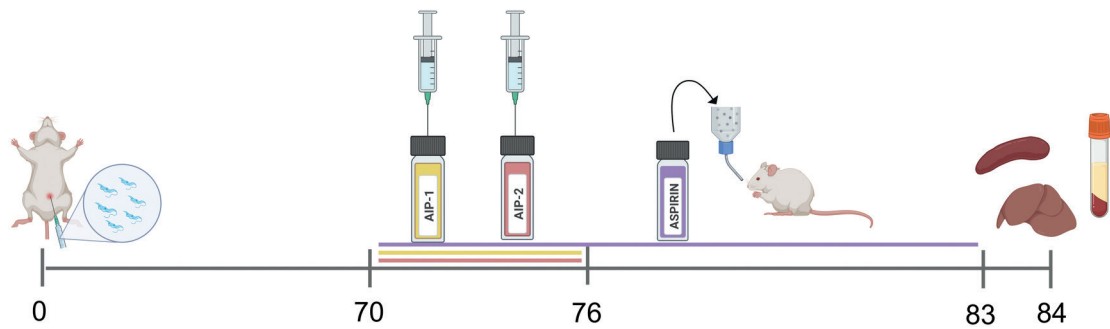

Fig. 1: study timeline. At 84 days post-infection (dpi), liver, spleen, and serum samples were collected for analysis, including parasite quantification, histopathology, cytokine profiling, gene expression, flow cytometry, and serum enzyme levels. Image created with BioRender.com.

*DNA extraction and parasite burden* - DNA was extracted from a frozen liver section using PDQeX Nucleic Acid Extractor and PreGem Universal kits (MicroGem International) according to the manufacturer's protocol. Levels of *T. cruzi* in the liver were quantified using Taqman primers and probes for the *T. cruzi* satellite DNA and mouse GAPDH as previously described[24] [Supplementary data (Table I)]. Samples were amplified using a QuantStudio 7 Pro Real Time thermocycler. The observed Ct mean for each sample was normalized to calculate the parasite burden per mg tissue using the equation derived from the standard curve.

*Histopathology* - Formalin-fixed, paraffin-embedded liver tissues were sectioned at 5 μm and serial sections spaced 10-30 μm apart were analyzed to represent approximately one-quarter of the organ as described previously.[25] Sections were stained with H&E for inflammation and Picrosirius Red for fibrosis. For each section, four representative fields were captured at 10× magnification using an AmScope 40X-2000X microscope. Inflammation was quantified by counting lymphocyte nuclei in Fiji ImageJ, applying a particle size filter to exclude hepatocyte nuclei and debris, with counts from the four selected fields averaged per section. Fibrotic areas were measured as pink-stained regions, and all values were normalized to tissue area to enable accurate comparisons across sections and animals.

*Preparation of liver lysate* - A snap-frozen liver section was maintained on ice and transferred to a GentleMACS M-tube containing RIPA Lysis and Extraction Buffer (Thermo Fisher) to ensure protein integrity and minimize degradation during subsequent processing. Subsequently, the liver was homogenized using the gentleMACS™ Dissociator's Protein saved program. The resulting lysate was centrifuged at 500 x g for 5 min, and the supernatant was carefully transferred to a low-temperature resistant Eppendorf tube. Total protein concentration was determined using Pierce™ BCA Protein Assay Kits (Thermo Fisher). Stock lysate was stored at -80ºC until use.

*Luminex multiplex bead panel* - The liver lysate was diluted to 5 μg/μL for the subsequent assessment of cytokines and proteins using Luminex assays. Two kits were used for these assays: the MILLIPLEX® Phospho/Total STAT3 Magnetic Bead 2-Plex Kit (Millipore Sigma) and the Mouse Th17 Panel MAGNETIC with IL-6, IFN-γ, TNF-α, IL-2, IL-4, and IL-10 (Millipore Sigma). Protein and cytokine levels were then determined and normalized to the standards provided with the respective kits.

*RNA extraction and reverse transcription polymerase chain reaction (RT-PCR)* - RNA was extracted from a frozen liver section using the Qiagen RNeasy Plus Mini Kit, following the manufacturer's protocol. The RNA concentration was determined using the Thermo Fisher NanoDrop One/OneC UV-Vis Spectrophotometer, and the final product was diluted to 10 ng/μL with RNase-free water for RT-PCR. An RT-PCR mix was prepared using the High-Capacity cDNA Reverse Transcription Kit (Applied Biosystems™), which was then added to each diluted sample before using the Biorad T100 Thermal Cycler.

For RT-PCR, the thermal cycler was programmed to maintain 25ºC for 10 min, followed by 37ºC for 2 h, then 85ºC for 5 min, and finally held at 4ºC until the

tubes were retrieved. cDNA was stored in -20ºC until use. The cDNA was diluted 1:10 to evaluate gene expression using quantitative PCR (QuantStudio 7 Pro). Gene expression analysis was performed using TaqMan primers targeting Nuclear factor kappa-light-chain-enhancer of activated B cells *(Nfk-B)*, Signal transducer and activator of transcription 1 *(Stat-1)*, Nitric oxide synthase 2 *(Nos2)*, Cyclooxygenase-2 *(Cox2)*, Arginase 1 *(Arg1)*, Matrix metalloproteinase-9 *(Mmp9)*, B-cell translocation gene *(Btg2)*, and Nuclear factor erythroid 2-related factor 2 *(Nfe2I2)* [Supplementary data (Table II)]. Each sample was analyzed in duplicate. The relative expression level (RQ) of each target gene was determined using the 2-ΔΔCt method, where ΔCt is the difference between the Ct values of the target gene and *Gapdh* [Ct(target) - Ct(*Gapdh*)], and ΔΔCt is the difference between the ΔCt of the sample and the average ΔCt of the uninfected untreated control mice. The RQ was then calculated using the formula RQ = 2(-ΔΔCt).[20]

*Single cell suspension of splenocytes* - Spleens were first rinsed with sterile 1X PBS and transferred to a gentleMACS C Tube containing 3 mL of sterile PBS for homogenization using a gentleMACS Dissociator (Miltenyi Biotech). To lyse red blood cells, ACK lysis buffer (Lonza) was added to the spleen homogenates, and the solution was diluted 5-fold with RPMI medium supplemented with 10% fetal bovine serum (FBS), 1X penicillin-streptomycin (Pen-Strep), and L-glutamine (cRPMI medium). Splenocytes were then pelleted by centrifugation at 300×g for 5 min. The resulting pellet was resuspended in 5 mL of cRPMI medium and passed through a 40 μm filter (BD Biosciences) to remove debris. Cell viability was assessed using acridine orange-propidium iodide (AOPI) live/dead dye and a Cellometer Auto 2000 (Nexcelom Bioscience) automated cell counter. For each sample, $1 \times 10^6$ viable splenocytes were plated in a 96-well non-tissue culture plate and cells were stained for immunophenotyping analysis.

*Immunophenotyping panel* - Immunophenotyping was performed by first staining $1 \times 10^6$ splenocytes with ViaDye Red fixable viability dye for 30 min on ice. After viability assessment, cells were incubated with a cocktail of fluorochrome-conjugated antibodies targeting surface markers - CD3, CD4, CD8, CD25, CD11b, CD11c+, Ly6G and Ly6C – [Supplementary data (Table III)] for 30 min on ice in the dark. Then, it was fixed using BD Cytofix for 20 min. After fixation, cells were resuspended and analyzed using an Aurora spectral flow cytometer. Data analysis was performed using FlowJo software. The representative gating strategy is shown in Supplementary data (Figure).

*Liver enzymes* - Whole blood was collected in serum separator tubes and left to clot at room temperature (RT) for at least 30 min. The tubes were then centrifuged at 10,000xg for 5 min at RT to separate the serum. The serum was transferred to a cryovial and frozen at -80ºC. A 100 μL serum sample was sent to the Clinical Pathology Laboratory at Baylor College of Medicine for analysis. Liver enzymes, including albumin (ALB), total protein (TP), alanine transaminase (ALT), aspartate transaminase (AST), lactate dehydrogenase (LDH), alkaline phosphatase (ALP), direct bilirubin (DBILC), and total bilirubin (TBILC), were measured.

*Statistical analysis* - All data were analyzed and visualized using GraphPad Prism (Version 10.2.3). Statistical significance was determined by two-way analysis of variance (ANOVA) with multiple comparisons relative to the Infected Untreated control group (p < 0.05). In addition, liver enzyme levels from serum samples were also compared to the Uninfected Untreated control using the same analysis. Error bars represent the mean ± standard deviation (SD). Significance is indicated as follows: *p ≤ 0.05; **p ≤ 0.01; ***p ≤ 0.001; ****p ≤ 0.0001.

## RESULTS

*Parasite load, hepatic cellular infiltration, and fibrotic changes* - Our results confirm successful infection in the mice, as demonstrated by an increased parasite burden across all infected groups compared to the Uninfected Untreated group (Fig. 2A). Although no statistically significant differences in liver parasite burdens were observed, notable variations in cellular infiltrate and fibrosis were detected. As expected, infection led to a significant increase in hepatic cellular infiltrate compared to uninfected mice. Similar to our previous cardiac study,[20] parasitemia levels and cardiac parasite burdens did not differ with treatment, indicating that the observed effects are independent of parasite load. Treatment with AIP-1, AIP-2, and aspirin each resulted in a further significant increase in cellular infiltrate compared to the Infected Untreated group (Fig. 2B).

Additionally, infection induced a significant increase in fibrosis relative to Uninfected Untreated mice; however, only aspirin treatment significantly reduced fibrosis when compared to Infected Untreated mice (Fig. 2C). These data suggest that the immunomodulatory effects of AIP-1 and AIP-2 in the liver are not mediated by reductions in parasite burden.

Interestingly, while fibrosis levels remained unchanged in the AIP-1- and AIP-2-treated groups compared to the Infected Untreated group, cellular infiltration followed a distinct pattern. These findings highlight the complex relationship between immune response and liver pathology in *T. cruzi* infection, underscoring the need for further investigation into how AIP-1 and AIP-2 influence these processes.

*Treatments with AIP-1 and AIP-2 promote a balanced hepatic immune response* - To assess the impact of treatments on inflammatory cytokines in the liver, Luminex assays were performed on liver protein extracts. Pro-inflammatory cytokines — including IL-6, IFN-γ, and TNF-α — were measured alongside anti-inflammatory markers such as IL-2, IL-4, and IL-10.

Our results show that, compared to Uninfected Untreated mice, infection alone led to a significant decrease in IL-6 (Fig. 3A), IFN-γ (Fig. 3B), and IL-10 (Fig. 3F), with no significant differences observed in TNF-α (Fig. 3C), IL-2 (Fig. 3D), or IL-4 (Fig. 3E). Interestingly, treatment with AIP-1 and AIP-2 resulted in a significant

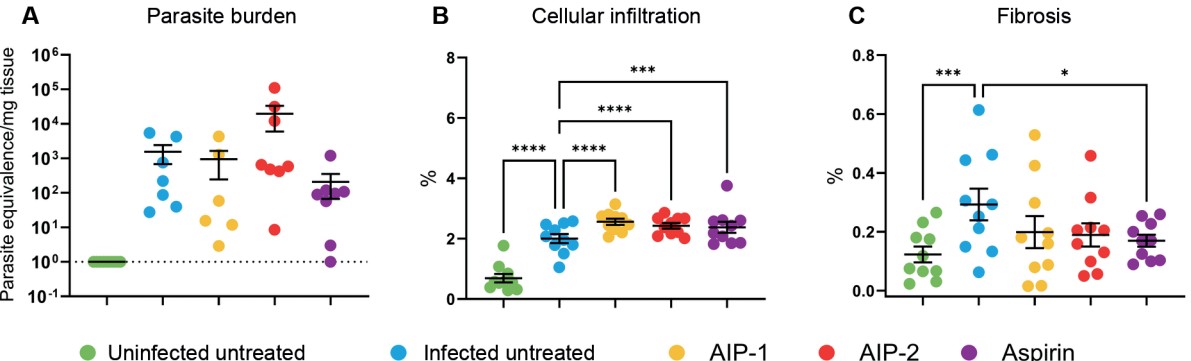

Fig. 2: parasite burden and tissue pathology. Parasite burden in the liver was measured at the end of the study by quantitative polymerase chain reaction (qPCR). Hepatic cellular infiltration was evaluated using Hematosin & Eosin (H&E) staining, while fibrosis was assessed through Picrosirius Red staining. The resulting images were analyzed with ImageJ software. (A) Parasite burden; (B) Cellular infiltration; (C) Fibrosis. Data from individual mice are shown, n = 10. Error bars are defined by mean with standard deviation (SD). *p ≤ 0.05; ***p ≤ 0.001; ****p ≤ 0.0001.

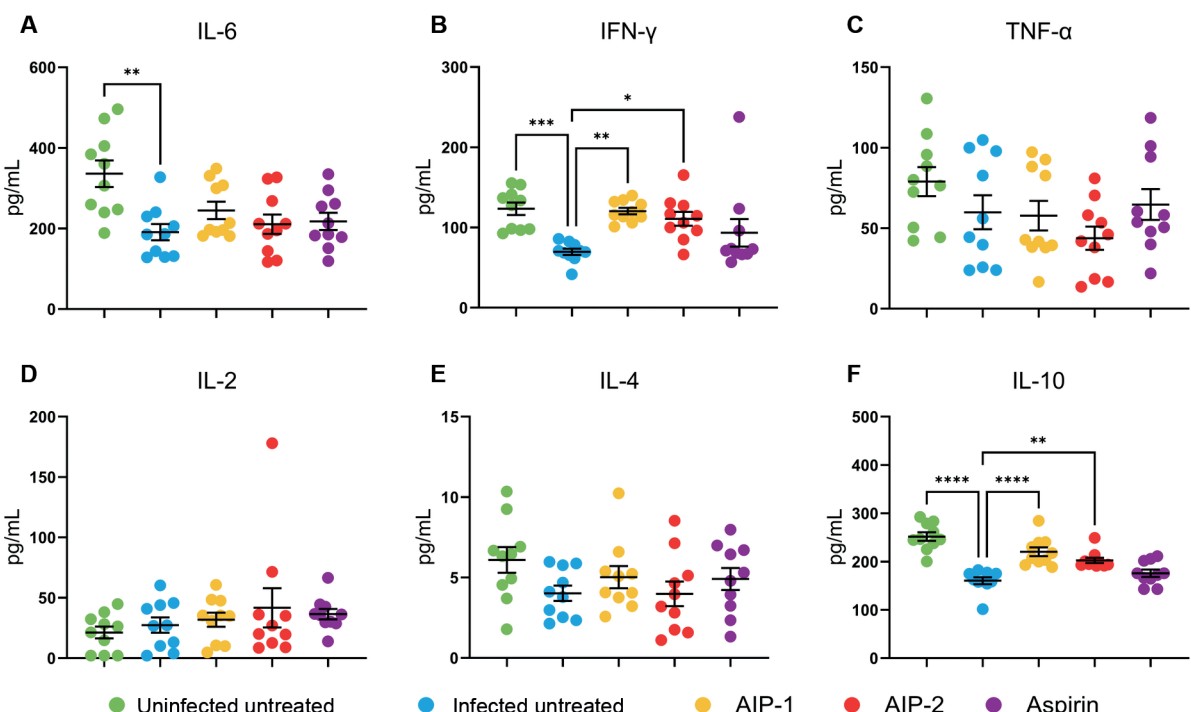

Fig. 3: cytokines from liver lysate. Pro- and anti-inflammatory cytokine levels were quantified using protein extracts from liver samples. (A) Interleukin (IL)-6 ; (B) Interferon (IFN)-γ; (C) Tumor necrosis factor (TNF)-α; (D) IL-2; (E) IL-4; (F) IL-10. Data from individual mice are shown, n = 10. Error bars are defined by mean with standard deviation (SD). *p ≤ 0.05; **p ≤ 0.01; ***p ≤ 0.001; ****p ≤ 0.0001.

increase in IFN-γ (Fig. 3B) and IL-10 (Fig. 3F) compared to the Infected Untreated group. No significant changes were observed in IL-6, TNF-α, IL-2 or IL-4 levels following AIP-1 or AIP-2 treatment.

Together, these data suggest that AIP-1 and AIP-2 may promote a more balanced $T_H1/T_H2$ hepatic immune environment, characterized by elevated levels of both pro- and anti-inflammatory cytokines. In our previous study,[20] we reported that serum cytokine levels differed from cardiac tissue cytokine levels, and here we similarly show that hepatic cytokine levels follow a distinct pattern from serum. This underscores the importance of examining local tissue immune responses to fully assess the impact of treatment.

*Treatment with AIP-1 reduced levels of pSTAT3* - Our findings show no significant differences in pSTAT3 or total STAT3 levels among the treatment groups compared to the Infected Untreated group (Fig. 4A-B). However, *T. cruzi* infection alone significantly increases the pSTAT3:STAT3 ratio relative to Uninfected Untreated mice, while AIP-1 treatment trends toward reducing this ratio, with a p-value of 0.0761 (Fig. 4C).

*Treatment with AIP-1 and AIP-2 modulate expression of inflammatory and oxidative stress genes - T. cruzi* infection induces inflammation and oxidative stress in the liver.[16,26] In our model, infected untreated mice showed increased expression of *Nfκ-B* (Fig. 5A) and *Stat-1* (Fig. 5B) compared to Uninfected Untreated mice, while the expression of *Arg1* (Fig. 5C), *Mmp9* (Fig. 5D), *Btg2* (Fig. 5E), and *Nfe2l2* (Fig. 5F) was decreased. Treatment with AIP-1, AIP-2, and aspirin reduced the expression of *Nfκ-B* and *Stat-1* relative to Infected Untreated mice, suggesting that these treatments downregulate inflammatory gene expression.

Additionally, AIP-1 and AIP-2 significantly increased the expression of *Arg1* (Fig. 5C) and *Nos2* (Fig. 5G) compared to Infected Untreated mice. Co-expression of *Arg1* and *Nos2* is a hallmark of myeloid-derived suppressor cells (MDSCs), which play a role in nitric oxide-mediated suppression of immune cell proliferation and promote survival in experimental infection models.[26] These findings suggest that AIP-1 and AIP-2 may enhance recruitment of MDSCs to the liver, thereby reducing inflammation.

Interestingly, only AIP-1 treatment increased *Mmp9* expression compared to Infected Untreated mice (Fig. 5D). *Mmp9* is involved in tissue remodeling and fibrosis during CD,[27] but it has also been shown to release TNF-α from cell surfaces, indicating a dual role in promoting inflammation.[28] Our observation of reduced *Mmp9* expression in Infected Untreated mice compared to the Uninfected control suggests that *T. cruzi* may suppress *Mmp9* as an immune evasion strategy, which is potentially reversed by AIP-1 treatment.

Finally, AIP-1 significantly increased *Btg2* expression relative to Infected Untreated mice (Fig. 5E). *Btg2* is known to trigger antioxidant responses, including activation of *Nfe2l2*, and is typically downregulated in oxidative stress and cancer models.[29] The *Nfe2l2* pathway is a key antioxidant defense mechanism in the liver.[17] In this study, we show that *T. cruzi* infection significantly reduces the expression of *Btg2* and *Nfe2l2*, while treatment with AIP-1 and AIP-2 trends toward restoring their expression (Fig. 5E-F), suggesting a potential role in enhancing protective antioxidant responses during infection.[30,31]

*Alterations in splenic immune cell populations following T. cruzi infection and treatment with AIP-1 and AIP-2* - Splenocyte single-cell suspensions were stained with CD3, CD11b, CD11c, Ly6G, Ly6C, CD4, CD8, and CD25, and analyzed by flow cytometry. All infected mice exhibited a decrease in the CD3+CD25+ population compared to Uninfected Untreated control, with a significant reduction observed in the Infected Untreated group (Fig. 6A). While the CD4+ population was reduced across all infected groups, the CD8+ population increased in response to infection. Notably, mice treated with AIP-1, AIP-2, or aspirin exhibited a trend toward reduced frequencies of CD4+ T cells accompanied by a corresponding increase in CD8+ T cells (Fig. 6B-C), suggesting a potential shift in T cell dynamics.

We also analyzed monocyte populations and observed that treatment with AIP-1 and AIP-2 led to significant alterations in splenic immune cell composition. AIP-1 notably increased the frequency of CD11b+CD11c+ dendritic cells (DCs) and CD11b+Ly6GloLy6C+ MDSCs. Specifically, infected mice treated with AIP-1 displayed a significant rise in both DCs (Fig. 6D) and MDSCs (Fig. 6E) compared to Infected Untreated mice, suggesting that AIP-1 promotes the expansion of these regulatory immune cell populations.

In contrast, no significant differences were observed in the frequencies of CD11b+Ly6G+Ly6C− neutrophils (Fig. 6F) or CD4+CD25+ regulatory T cells (Fig. 6G) across treatment groups relative to the Infected Untreated control.

*Liver enzymes suggest a favorable response from the treatment groups* - Plasma samples were collected and analyzed to assess liver function and identify potential abnormalities resulting from infection or treatment.

ALB and TP levels serve as indicators of liver synthetic function, with reduced ALB often linked to chronic liver disease or inflammation.[32,33] Treatment with AIP-2 resulted in lower ALB levels compared to the Infected Untreated group; however, no significant differences were observed between the treatment groups and the Un-

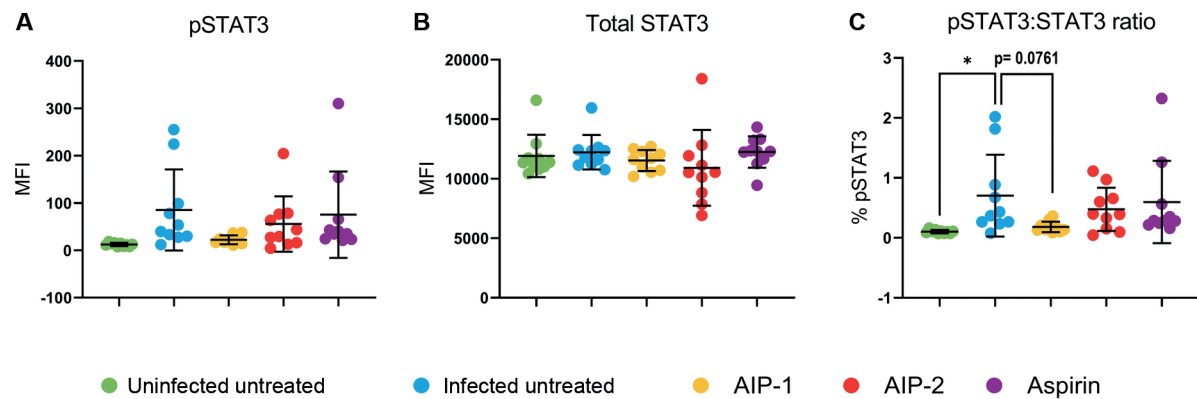

Fig. 4: signal transducer and activator of transcription 3 (STAT3) levels from liver lysate. Total STAT3 and phosphorylated (p)-STAT3 levels were quantified by Luminex assays. (A) pSTAT3 levels; (B) Total STAT3 levels; (C) pSTAT3 ratio. Data from individual mice are shown, n = 10. Error bars are defined by mean with standard deviation (SD). *p ≤ 0.05.

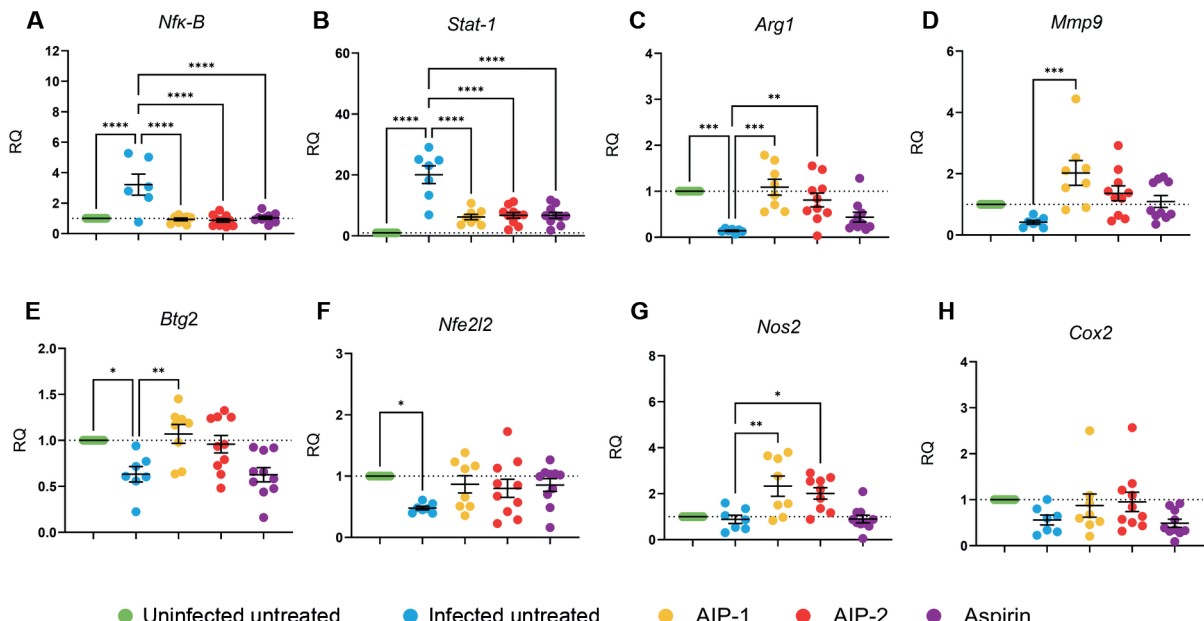

Fig. 5: relative quantification of gene expression levels in liver tissues. Gene expression was quantified using qPCR on cDNA synthesized from each liver sample. (A) Nuclear factor kappa-light-chain-enhancer of activated B cells (*Nfk-B*); (B) Signal transducer and activator of transcription 1 (*Stat-1*); (C) Arginase 1 (*Arg1*); (D) Matrix metalloproteinase-9 (*Mmp9*); (E) B-cell translocation gene 2 (*Btg2*); (F) Nuclear factor, erythroid 2-like 2 (*Nfe2l2*); (G) Nitric oxide synthase 2 (*Nos2*); (H) Cyclooxygenase-2 (*Cox2*). Data from individual mice are shown, n = 10. Error bars are defined by mean with standard deviation (SD). *p ≤ 0.05; **p ≤ 0.01; ***p ≤ 0.001; ****p ≤ 0.0001.

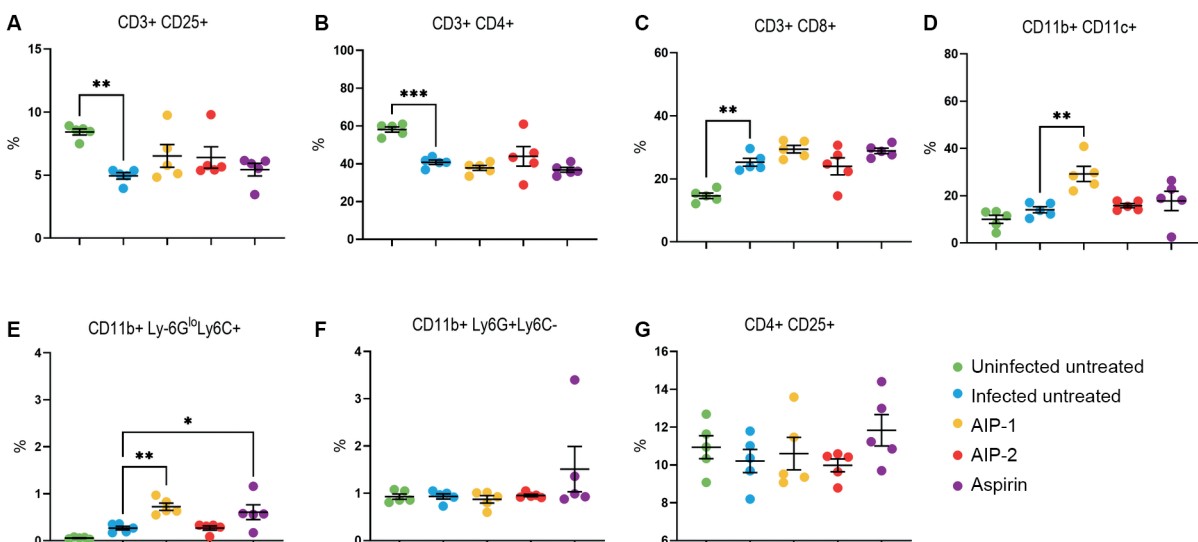

Fig. 6: immunophenotyping of immune cell populations. Splenocytes were stained and acquired by flow cytometry. (A) Cluster of differentiation (CD)3+CD25+; (B) CD3+CD4+; (C) CD3+CD8+; (D) CD11b+CD11c+; (E) CD11b+Ly6G^loLy6C+; (F) CD11b+Ly6G+Ly6C-; (G) CD4+CD25+. Data from individual mice are shown, n = 10. Error bars are defined by mean with standard deviation (SD). *p ≤ 0.05, **p ≤ 0.01, ***p ≤ 0.001.

infected Untreated control, except for a notable increase in ALB levels in the Aspirin group (Fig. 7A). Similarly, all infected groups showed elevated TP levels relative to the Uninfected Untreated control, suggesting that *T. cruzi* infection may impair liver synthetic function (Fig. 7B).

ALT and AST are key markers of hepatocellular damage and are commonly used to monitor treatment response, as elevations in these enzymes post-treatment can indicate hepatotoxicity in Chagasic patients.[10,33] In

this study, treatment with either AIP-1 or AIP-2 did not significantly affect ALT or AST levels when compared to either the Uninfected Untreated or Infected Untreated groups (Fig. 7C-D).

LDH, although a nonspecific marker, is often associated with hepatocellular damage.[34] Interestingly, LDH levels were significantly reduced in the AIP-1 and AIP-2 treatment groups compared to the Uninfected Untreated control (Fig. 7E). Lastly, ALP and DBILC are typically

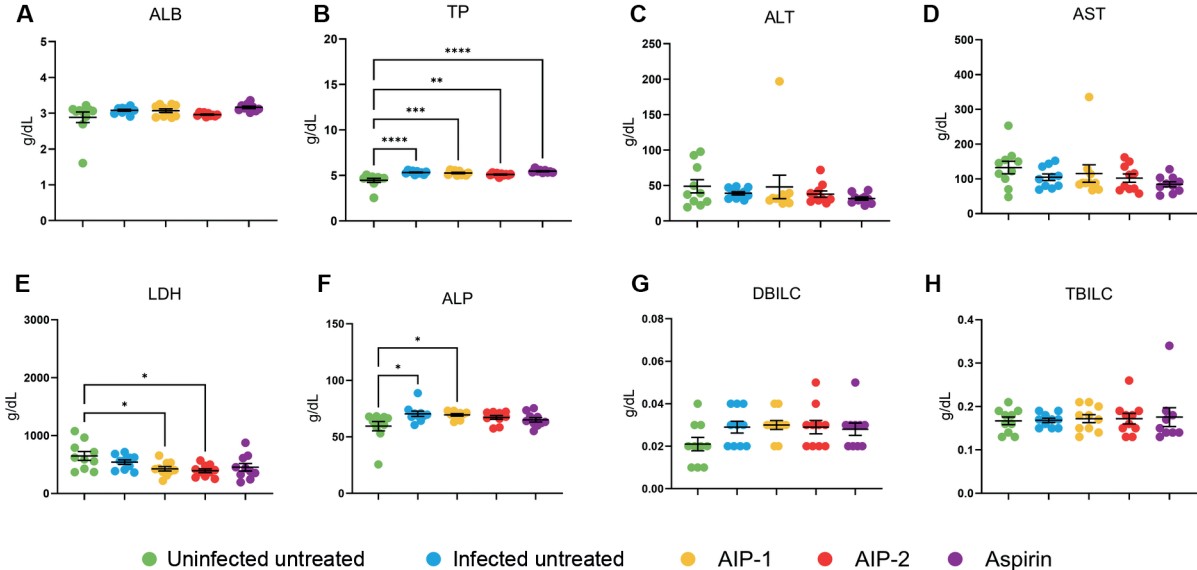

Fig. 7: liver enzymes. Whole blood plasma was used to quantify liver function enzymes to evaluate for clinical implications of abnormalities in the liver. (A) albumin (ALB); (B) total protein (TP); (C) alanine transaminase (ALT); (D) aspartate aminotransferase (AST); (E) lactate dehydrogenase (LDH); (F) alkaline phosphatase (ALP); (G) direct bilirubin (DBILC); (H) total bilirubin (TBILC). Data from individual mice are shown, n = 10. Error bars are defined by mean with standard deviation (SD). *$p \leq 0.05$, **$p \leq 0.01$, ***$p \leq 0.001$, ****$p \leq 0.0001$.

associated with biliary obstruction. The observed increase in ALP levels in the AIP-1 group compared to the Uninfected Untreated control may reflect infection-related changes (Fig. 7F).

## DISCUSSION

We previously demonstrated in our experimental mouse model of early chronic *T. cruzi* infection that the hookworm-derived proteins AIP-1 and AIP-2 significantly reduce cardiac cellular infiltrate and expression of inflammatory genes and cytokines.[20] Building on these findings, this study explored the liver-specific immune response to AIP-1 and AIP-2 treatment, as the liver plays a critical role in the clearance of blood-stage *T. cruzi* parasites and the induction of protective immune responses during chronic infection.[35,36] Given the significant inflammatory burden associated with *T. cruzi*-induced liver pathology, we aimed to determine whether AIP-1 and AIP-2 could mitigate inflammation while preserving hepatic function. Our findings suggest that these recombinant proteins function not merely as anti-inflammatory agents, but as immune modulators with potential antioxidant properties — capable of rebalancing immune responses, dampening pro-inflammatory signaling pathways, and promoting protective gene expression, all without causing hepatotoxicity or worsening fibrosis.

We observed increased cellularity in the liver (Fig. 2B), however levels of the proinflammatory cytokines IL-6 and TNF-α remained unchanged across infected treatment groups, suggesting that the increased immune cell presence was not inflammatory in nature. Instead, flow cytometry data revealed that AIP-1 treatment elevated the proportion of CD11b⁺CD11c⁺ dendritic cells and CD11b⁺Ly6G^lo Ly6C⁺ MDSCs (Fig. 6D-E). Given the well-established roles of MDSCs in dampening immune overactivation and promoting tissue repair[37,38] their expansion may explain the observed immune cell infiltration without an associated rise in proinflammatory cytokines. MDSCs are also known to inhibit T-cell proliferation and contribute to IL-10-mediated downregulation of hepatic stellate cell activation, potentially mitigating fibrosis.[38] Although we observed increased MDSCs in the spleen following AIP-1 treatment, further investigation is needed to determine whether similar populations accumulate in the liver and contribute to the increased infiltrate observed histologically in Fig. 2B.

Consistent with previous studies, *T. cruzi* infection increased hepatic pSTAT3 levels, a key driver of inflammation and fibrosis.[39,40,41] AIP-1 treatment was associated with a trending reduction in the pSTAT3 ratio (Fig. 4C), which may help explain the observed tendency toward reduced fibrosis in treated mice, despite the increase in immune cell infiltration. STAT3 inhibition, alongside modest increases in *Mmp9* — a matrix metalloprotease involved in extracellular matrix remodeling — suggests a potential mechanism for preserving liver architecture without promoting fibrogenesis.[42,43] Although STAT-3 is typically activated by IL-6, it can also be triggered by other cytokines, including IL-10.[44] In our previous cardiac study,[20] infection was associated with elevated serum IL-10, which may have contributed to the observed increase in hepatic pSTAT3, even though IL-10 levels within the liver were reduced by *T. cruzi* infection. Because both IL-6 and IL-10 can be produced by immune and non-immune cells in the liver, future studies will aim to characterize the liver immune cell composition to identify the primary cytokine-producing cells and determine how AIP-1 and AIP-2 affect both immune and non-immune cell populations.

Transcriptional profiling revealed further support for this immune-modulatory role. AIP-1 and AIP-2 treatments upregulated protective genes such as *Btg2, Nos2, Nfe2l2*, while downregulating proinflammatory mediators *Nfк-B* and *Stat-1* (Fig. 5). These changes highlight a shift toward a regulated immune state that may help maintain hepatic function in the face of persistent infection.

The roles of *Btg2*, *Nos2*, and *Nfe2l2* in liver health are particularly relevant.[15,45,46] Btg2 is associated with antioxidant defense and reduced cellular proliferation[30] while *Nfe2l2* orchestrates the hepatic antioxidant response and offers cytoprotective effects during oxidative stress.[46] *Nos2* typically associated with nitric oxide production and antimicrobial defense, has a complex role in CD. During acute infection, *Nos2* promotes parasite clearance and Th1 immune responses. However, in the chronic phase, sustained or excessive *Nos2* expression can contribute to tissue damage, immune dysregulation, and fibrosis.[45] In our study, we observed elevated *Nos2* expression in both AIP-1 and AIP-2 treatment groups (Fig. 5C), yet this was not accompanied by increased fibrosis or hepatocellular damage. These findings suggest that AIP treatments may modulate *Nos2* activity in a way that maintains host defense — possibly through oxidative stress regulation — while mitigating tissue injury.

Importantly, despite transcriptional upregulation of *Nos2* we did not observe a reduction in liver parasite burden. This finding suggests that AIP-1 and AIP-2 do not act as antiparasitic agents per se but instead modulate the immune environment—potentially supporting host tolerance rather than resistance. This distinction is critical; while parasite control remains a therapeutic goal, preventing tissue damage is important to preserve tissue function in chronic CD.[47]

Supporting this, neither AIP-1 nor AIP-2 increased liver enzymes (ALT, AST, LDH, or ALP), reinforcing their hepatoprotective profile (Fig. 7). These results contrast sharply with previous observations where curative doses of BNZ significantly elevated liver enzymes, indicating hepatotoxicity.[18,25] Thus, AIP-based therapies may offer a safer alternative or complement to antiparasitic treatments by mitigating tissue damage without compromising liver function.

While our findings suggest that AIP-1 and AIP-2 modulate immune responses without exacerbating liver pathology, several limitations remain. Further investigation is needed to determine whether the immune cells recruited from the liver are regulatory in nature. In particular, the role of MDSCs in the liver during *T. cruzi* infection requires clarification, as our current data are limited to splenic populations. STAT-1 protein and activation were not assessed, which will be important in future studies to fully understand IFN-γ-mediated signaling. Additional flow cytometry analyses are necessary to more precisely characterize the cellular composition of the hepatic infiltrate. Moreover, our study did not differentiate responses based on sex, and future comparisons between male and female mice will be important to assess potential sex-specific differences in immune modulation and treatment efficacy.

*Trypanosoma cruzi* infection is well known to trigger a robust pro-inflammatory response, often resulting in tissue damage and fibrosis, particularly within the liver.[35] In our model, infection led to elevated parasite burden, increased cellular infiltration, and pronounced liver fibrosis — hallmarks of CD pathology — confirming that our model effectively replicates the characteristic disease features (Fig. 2). Interestingly, treatment with AIP-1 and AIP-2 further increased cellular infiltration without significantly reducing fibrosis (Fig. 2B-C), suggesting that these proteins may primarily influence immune cell recruitment rather than directly modulating fibrotic processes. In our previously published study evaluating the anti-inflammatory effects of AIP-1 and AIP-2 in the heart,[20] we observed a significant reduction in cardiac inflammatory markers at 84 days post-infection, although only aspirin treatment significantly reduced cardiac fibrosis. Notably, we have also shown that cardiac fibrosis in this mouse model is progressive.[48] Therefore, potential anti-fibrotic effects of AIP treatment may not yet be evident at this time point. Inclusion of additional time points — both immediately after treatment and at later intervals — would allow a more comprehensive understanding of the kinetics of fibrosis and immunomodulation. We plan to address this in future studies through time-course experiments designed to capture both the short- and long-term effects of AIP treatment.

In summary, our findings indicate that AIP-1 and AIP-2 act as immune modulators with potential antioxidant properties, helping to preserve liver structure and function during chronic *T. cruzi* infection. They promote immune cell infiltration — likely of regulatory and suppressor cell types — without exacerbating fibrosis, possibly through downregulation of pSTAT3, *Stat1*, and *Nfк-B*, and upregulation of protective genes such as *Btg2*, *Mmp9*, *Nos2*, and *Nfe2l2*. Although these treatments do not reduce parasite burden, their capacity to fine-tune immune responses and prevent pathological inflammation highlights their potential as adjunctive therapies that could be combined with antiparasitic treatment to improve clinical outcomes in CD.

*Conclusions and future directions* - Overall, our findings suggest that AIP-1 and AIP-2 exert anti-inflammatory and antioxidant effects in chronic *T. cruzi* infection by reducing inflammatory signaling, promoting a balanced cytokine response, and preserving liver function. The data suggests that the immunomodulatory effects of may be mediated in part by regulatory DCs and MDSCs, which were increased in the spleen. Overall, this data provides promising insights into the therapeutic potential of these proteins. However, further studies are warranted to delineate their precise mechanisms of action and assess their long-term effects in chronic CD. Future research should also explore whether AIP-1 and AIP-2 can synergize with existing anti-parasitic therapies, such as BNZ, to enhance treatment efficacy and improve disease outcomes.

## ACKNOWLEDGEMENTS

To Liyan Chen and Gonteria J Robinson for their contributions to generating the reagents used in this study.

## AUTHORS' CONTRIBUTION

KJ and BZ designed the study; BZ generated reagents; MJV, MG and CP acquired the data; MJV and CP analyzed the data; MJV, CP, KJ drafted the manuscript. All authors contributed to the article and approved the submitted version. The authors declare no conflict of interest.

## DATA AVAILABILITY

The contents underlying the research text are included in the manuscript.

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

# OPEN PEER REVIEW

Memórias do IOC thanks the anonymous reviewers for their contribution to the peer review of this work.

**FIRST REVIEW ROUND**

REVIEWERS' COMMENTS

**REVIEWER #1**

The authors of the study investigated the effects of two hookworm proteins on hepatic fibrosis and immunomodulation in response to Trypanosoma cruzi infection. The effects of these proteins on liver histopathology, cytokines, parasite burden, and the expression of certain genes associated with oxidative stress and fibrosis were evaluated. Furthermore, the authors evaluated splenocytes and liver function through serum levels.

Line 49: I suggest "Splenocytes were assessed by flow cytometry..."

One curious aspect is the use of aspirin as the control. Could the authors explain the rationale behind choosing this medication, dosage, administration route, and time of treatment? Why didn't they use the saline solution used to wash and prepare the parasite inoculum.

The study design also raised some doubts: the AIP treatment was for seven days, but the authors only euthanized the groups nine days after the treatment ended. A critical question to address is whether the timing of the treatment and evaluation interval was adequate for detection of fibrosis and immunomodulation resulting from the treatment.

The representative gating strategy for the flow cytometry data was absent. A subset of the population exhibits a remarkably low percentage. What is the number of events acquired for each sample to evaluate these populations? How many cells were analyzed per animal?

Histopathological analysis was performed in only 4 representative images? This method should be better described: to evaluate this representative portion of the organ (1/4 of the liver), serial sections, 10-30 micrometers apart, should be analyzed in relation to cells count and tissue area of images.

The liver lysate was obtained with RIPA Buffer; however, the authors have to mention if they used phosphatase and protease inhibitors, which are essential for preserving the integrity of proteins for subsequent analysis, particularly for the detection of phosphorylated proteins.

The parasite burden was evaluated in the liver. Do the authors have the information of blood parasite count of this T. cruzi strain in this mouse model along time? This information could facilitate a more comprehensive understanding of the course and stage of infection. Furthermore, the systemic parasitemia could facilitate the comprehension of the immunological response manifested in the spleen and serve as a correlate with the hepatic parasite burden identified.

Similarly, the researchers had the option of measuring blood cytokines to create a correlation with the systemic data obtained from the spleen.

The authors conducted an evaluation of the total and activated STAT-3 protein, which is traditionally activated by IL-6. However, the present model demonstrated a decrease in IL-6 levels in response to the infection. Moreover, IFN-g exhibited diminished levels during infection, yet these levels were restored following treatment with both AIP-1 and AIP-2. The researchers' didn't evaluate STAT-1 protein, which is known to be activated in response to IFN-g. It would be interesting whether this approach will be further explored in subsequent studies. The STAT-1 gene expression does not necessarily reflect protein levels or activation.

The Ref [24] did not described GAPDH primers. Instead of that reference, I suggest the following https://doi.org/10.1038/emi.2013.50.

AUTHORS' RESPONSE TO THE REVIEWERS

On behalf of our co-authors, we thank the reviewers for their critique of our manuscript entitled "Therapeutic Potential of Hookworm Proteins in Promoting Regulatory Immune Responses to Modulate Trypanosoma cruzi-Induced Liver Inflammation and Oxidative Stress" for consideration for publication in Memórias do Instituto Oswaldo Cruz.

Below is a point by point response to each comment:

Reviewer: 1

Reviewer comments: The authors of the study investigated the effects of two hookworm proteins on hepatic fibrosis and immunomodulation in response to Trypanosoma cruzi infection. The effects of these proteins on liver histopathology, cytokines, parasite burden, and the expression of certain genes associated with oxidative stress and fibrosis were evaluated. Furthermore, the authors evaluated splenocytes and liver function through serum levels.

Line 49: I suggest "Splenocytes were assessed by flow cytometry..."

Response: Thank you for the suggestion; we have corrected this line in the manuscript.

One curious aspect is the use of aspirin as the control. Could the authors explain the rationale behind choosing this medication, dosage, administration route, and time of treatment? Why didn't they use the saline solution used to wash and prepare the parasite inoculum.

Response: This study builds on our previous work demonstrating the anti-inflammatory effects of hookworm-derived proteins in the heart. Since aspirin also has well-established cardiac anti-inflammatory effects and has shown to reduce cardiac inflammation in mouse models of T. cruzi infection, we used 25mg/kg aspirin in drinking water daily for 14 days to be consistent with Molina-Berrios, et al 2013 (doi: 10.1371/journal.pntd.0002173). Untreated T. cruzi-infected mice provided a baseline of inflammation, while aspirin served as a positive control with known mechanism of action for anti-inflammatory activity.

We have clarified this rationale in the revised manuscript under the "Experimental infection and treatments" section.

The study design also raised some doubts: the AIP treatment was for seven days, but the authors only euthanized the groups nine days after the treatment ended. A critical question to address is whether the timing of the treatment and evaluation interval was adequate for detection of fibrosis and immunomodulation resulting from the treatment.

Response: We appreciate the reviewer's thoughtful comment regarding the timing of treatment and evaluation. In the study by Cuellar et al (doi: 10.1371/journal.pntd.0000439) naive mice were treated with 50μg of recombinant AIP-1 by IP injection every 2 days for 8 days, and in a mouse model of asthma (doi: 10.1126/scitranslmed.aaf8807), mice were treated with 1mg/kg AIP-2 once daily for 5 days prior to OVA challenge. In considering these varied study designs, we elected to use once daily treatment of mice with 1mg/kg AIP-1 or AIP-2 for 7 days during the early chronic phase of T. cruzi infection in our established model. Aspirin treatment was administered for 14 days, thus we elected to euthanize all mice at 84 dpi, which corresponded to the end of Aspirin treatment, but approximately 7 days after treatment with AIP-1 and AIP-2 ended to determine if the anti-inflammatory effects were sustained after the end of treatment.

In our previously published study evaluating the anti-inflammatory effects of AIP-1 and AIP-2 in the heart (doi: 10.3389/fpara.2023.1244604), we showed that at 84 days post infection there was significant reduction in cardiac inflammatory markers, but only Aspirin treatment significantly reduced cardiac fibrosis. However, we have previously shown that in our mouse model, cardiac fibrosis is progressive (doi: 10.1161/JAHA.119.013365). Thus, it is possible that any potential anti-fibrotic effects may not yet be fully evident at this time point. Including additional time points—both immediately after treatment and at later intervals—would provide a more complete understanding of the kinetics of fibrosis and immunomodulation. We plan to address this in future studies through time-course experiments to capture both the short- and long-term impact of AIP treatment.

This was noted in the manuscript.

The representative gating strategy for the flow cytometry data was absent. A subset of the population exhibits a remarkably low percentage. What is the number of events acquired for each sample to evaluate these populations? How many cells were analyzed per animal?

Response: We appreciate this observation. On average, 182,148 events were acquired per sample (SEM = 11,738). We generated a supplementary figure 1 and it has been noted in the Results section.

Histopathological analysis was performed in only 4 representative images? This method should be better described: to evaluate this representative portion of the organ (1/4 of the liver), serial sections, 10-30 micrometers apart, should be analyzed in relation to cells count and tissue area of images.

Response: We appreciate the reviewer's suggestion. The manuscript has been revised to clarify that histopathological analysis was performed on serial sections representing approximately one-quarter of the liver. While this approach provides a representative evaluation of inflammation and fibrosis and a small, supporting dataset for the study, we acknowledge that, given the liver's large size, evaluating only a portion may not capture the full heterogeneity of the tissue. In future studies, we plan to analyze multiple liver sections from each animal to provide a more comprehensive assessment.

The liver lysate was obtained with RIPA Buffer; however, the authors have to mention if they used phosphatase and protease inhibitors, which are essential for preserving the integrity of proteins for subsequent analysis, particularly for the detection of phosphorylated proteins.

Response: RIPA buffer was used to maintain consistency with the cardiac evaluation of these proteins, and phosphatase and protease inhibitors were not included in this study. However, all the liver sections were snap-frozen immediately upon collection and used promptly for protein isolation, which likely prevented detectable protein degradation and did not interfere with the assays performed. We appreciate the reviewer's suggestion and will incorporate phosphatase and protease inhibitors in future studies to enhance sensitivity and optimize results.

This has been clarified to indicate that tissues were snap-frozen and processed immediately for the assays in the Methods section for liver collection.

The parasite burden was evaluated in the liver. Do the authors have the information of blood parasite count of this T. cruzi strain in this mouse model along time? This information could facilitate a more comprehensive understanding of the course and stage of infection.

Furthermore, the systemic parasitemia could facilitate the comprehension of the immunological response manifested in the spleen and serve as a correlate with the hepatic parasite burden identified.

Response: We thank the reviewer for this suggestion. Since this study is a follow-up to our recently published cardiac evaluation, parasitemia data as well as cardiac parasite burdens for this T. cruzi strain in this mouse model are described in the previous paper (doi: 10.3389/fpara.2023.1244604). We found that parasitemia levels and cardiac parasite burdens were not significantly different between treatment groups. Thus, we concluded that the immunologic differences between AIP-1/AIP-2 treated groups, and infected untreated controls were due to the direct immunomodulatory effects of these proteins rather than indirect effects that alter parasite levels.

This is noted in the Results section - "Parasite Load, Hepatic Cellular Infiltration, and Fibrotic Changes".

Similarly, the researchers had the option of measuring blood cytokines to create a correlation with the systemic data obtained from the spleen.

Response: As part of our cardiac evaluation, we assessed systemic cytokine levels in the serum and found that infected untreated mice exhibited elevated IFN-γ and IL-10 and reduced IL-2. Treatment with AIP-1 restored IL-2 levels without affecting the other cytokines. We also measured serum IgG levels specific for AIP-1 and AIP-2. This data is presented in Figure 5 of our previous publication (doi: 10.3389/fpara.2023.1244604).

This is noted in the manuscript.

The authors conducted an evaluation of the total and activated STAT-3 protein, which is traditionally activated by IL-6. However, the present model demonstrated a decrease in IL-6 levels in response to the infection.

Response: We thank the reviewer for this comment. While STAT-3 is classically activated by IL-6, it can also be triggered by other cytokines such as IL-10 (doi: 10.3389/fimmu.2023.1160719). In our previous cardiac study, we did see elevated levels of IL-10 in the serum with infection, which could have contributed to increased pSTAT3 levels in the liver, despite liver tissue levels of IL-10 being reduced by T. cruzi infection. Since IL-6 and IL-10 can be produced by both immune and non-immune cells within the liver, future studies will identify the composition of immune cells within the liver to better define the cytokine producing cells and the impact of AIP-1 and AIP-2 on immune cells and non-immune cells within the liver.

This is noted in the Discussion section.

Moreover, IFN-g exhibited diminished levels during infection, yet these levels were restored following treatment with both AIP-1 and AIP-2. The researchers didn't evaluate STAT-1 protein, which is known to be activated in response to IFN-g. It would be interesting whether this approach will be further explored in subsequent studies. The STAT-1 gene expression does not necessarily reflect protein levels or activation.

Response: We thank the reviewer for this comment. While IFN-γ levels were restored by AIP-1 and AIP-2, STAT-1 protein and activation were not assessed in this study. We acknowledge that gene expression alone may not reflect STAT-1 activity, and future studies should evaluate STAT-1 and phosphorylated STAT-1 to better understand IFN-γ–mediated signaling in response to these treatments.

This limitation has now been noted in the manuscript.

The Ref [24] did not describe GAPDH primers. Instead of that reference, I suggest the following https://doi.org/10.1038/emi.2013.50.

Response: We thank the reviewer for the suggestion; the reference has been updated as suggested.

In addition to this response letter, we are submitting the revised manuscript and supplemental data file with the changes highlighted for review. We feel that the revisions to the manuscript based on reviewer critique have greatly improved the manuscript. We thank you for your continued review and welcome any additional comments or critiques as you consider this manuscript for publication.

Sincerely,
Maria Jose Villar, MSc
National School of Tropical Medicine
Baylor College of Medicine
Texas Children's Hospital Center for Vaccine Development
Kathryn M. Jones, DVM, PhD
National School of Tropical Medicine
Baylor College of Medicine
Texas Children's Hospital Center for Vaccine Development

## SECOND REVIEW ROUND

### REVIEWERS' COMMENTS

### REVIEWER #1

The manuscript is very interesting and is ready to be published. The abstract, methodology, results, discussion and references are adequate. These results will help further research on this field.

