## [Reviewer Report · FIRST REVIEW ROUND - REVIEWERS COMMENTS]

## REVIEWER #1

The authors of the study investigated the effects of two hookworm proteins on hepatic fibrosis and immunomodulation in response to Trypanosoma cruzi infection. The effects of these proteins on liver histopathology, cytokines, parasite burden, and the expression of certain genes associated with oxidative stress and fibrosis were evaluated. Furthermore, the authors evaluated splenocytes and liver function through serum levels. 

Line 49: I suggest “Splenocytes were assessed by flow cytometry...”

One curious aspect is the use of aspirin as the control. Could the authors explain the rationale behind choosing this medication, dosage, administration route, and time of treatment? Why didn’t they use the saline solution used to wash and prepare the parasite inoculum.

The study design also raised some doubts: the AIP treatment was for seven days, but the authors only euthanatized the groups nine days after the treatment ended. A critical question to address is whether the timing of the treatment and evaluation interval was adequate for detection of fibrosis and immunomodulation resulting from the treatment.

The representative gating strategy for the flow cytometry data was absent. A subset of the population exhibits a remarkably low percentage. What is the number of events acquired for each sample to evaluate these populations? How many cells were analyzed per animal? 

Histopathological analysis was performed in only 4 representative images? This method should be better described: to evaluate this representative portion of the organ (1/4 of the liver), serial sections, 10-30 micrometers apart, should be analyzed in relation to cells count and tissue area of images. 

The liver lysate was obtained with RIPA Buffer; however, the authors have to mention if they used phosphatase and protease inhibitors, which are essential for preserving the integrity of proteins for subsequent analysis, particularly for the detection of phosphorylated proteins. 

The parasite burden was evaluated in the liver. Do the authors have the information of blood parasite count of this T. cruzi strain in this mouse model along time? This information could facilitate a more comprehensive understanding of the course and stage of infection. Furthermore, the systemic parasitemia could facilitate the comprehension of the immunological response manifested in the spleen and serve as a correlate with the hepatic parasite burden identified. 

Similarly, the researchers had the option of measuring blood cytokines to create a correlation with the systemic data obtained from the spleen.

The authors conducted an evaluation of the total and activated STAT-3 protein, which is traditionally activated by IL-6. However, the present model demonstrated a decrease in IL-6 levels in response to the infection. Moreover, IFN-g exhibited diminished levels during infection, yet these levels were restored following treatment with both AIP-1 and AIP-2. The researchers’ didn’t evaluate STAT-1 protein, which is known to be activated in response to IFN-g. It would be interesting whether this approach will be further explored in subsequent studies. The STAT-1 gene expression does not necessarily reflect protein levels or activation.

The Ref [24] did not described GAPDH primers. Instead of that reference, I suggest the following https://doi.org/10.1038/emi.2013.50.

---

## [Author Response · AUTHORS RESPONSE TO THE REVIEWERS]

## AUTHORS RESPONSE TO THE REVIEWERS

On behalf of our co-authors, we thank the reviewers for their critique of our manuscript entitled “Therapeutic Potential of Hookworm Proteins in Promoting Regulatory Immune Responses to Modulate Trypanosoma cruzi-Induced Liver Inflammation and Oxidative Stress” for consideration for publication in Memórias do Instituto Oswaldo Cruz.

Below is a point by point response to each comment:

## Reviewer: 1

Reviewer comments: The authors of the study investigated the effects of two hookworm proteins on hepatic fibrosis and immunomodulation in response to Trypanosoma cruzi infection. The effects of these proteins on liver histopathology, cytokines, parasite burden, and the expression of certain genes associated with oxidative stress and fibrosis were evaluated. Furthermore, the authors evaluated splenocytes and liver function through serum levels.

Line 49: I suggest “Splenocytes were assessed by flow cytometry...”

Response: Thank you for the suggestion; we have corrected this line in the manuscript.

One curious aspect is the use of aspirin as the control. Could the authors explain the rationale behind choosing this medication, dosage, administration route, and time of treatment? Why didn’t they use the saline solution used to wash and prepare the parasite inoculum.

Response: This study builds on our previous work demonstrating the anti-inflammatory effects of hookworm-derived proteins in the heart. Since aspirin also has well-established cardiac anti-inflammatory effects and has shown to reduce cardiac inflammation in mouse models of T. cruzi infection, we used 25mg/kg aspirin in drinking water daily for 14 days to be consistent with Molina-Berrios, et al 2013 (doi: 10.1371/journal.pntd.0002173). Untreated T. cruzi-infected mice provided a baseline of inflammation, while aspirin served as a positive control with known mechanism of action for anti-inflammatory activity. 

We have clarified this rationale in the revised manuscript under the “Experimental infection and treatments” section.

The study design also raised some doubts: the AIP treatment was for seven days, but the authors only euthanized the groups nine days after the treatment ended. A critical question to address is whether the timing of the treatment and evaluation interval was adequate for detection of fibrosis and immunomodulation resulting from the treatment.

Response: We appreciate the reviewer’s thoughtful comment regarding the timing of treatment and evaluation. In the study by Cuellar et al (doi: 10.1371/journal.pntd.0000439) naive mice were treated with 50µg of recombinant AIP-1 by IP injection every 2 days for 8 days, and in a mouse model of asthma (doi: 10.1126/scitranslmed.aaf8807), mice were treated with 1mg/kg AIP-2 once daily for 5 days prior to OVA challenge. In considering these varied study designs, we elected to use once daily treatment of mice with 1mg/kg AIP-1 or AIP-2 for 7 days during the early chronic phase of T. cruzi infection in our established model. Aspirin treatment was administered for 14 days, thus we elected to euthanize all mice at 84 dpi, which corresponded to the end of Aspirin treatment, but approximately 7 days after treatment with AIP-1 and AIP-2 ended to determine if the anti-inflammatory effects were sustained after the end of treatment.

In our previously published study evaluating the anti-inflammatory effects of AIP-1 and AIP-2 in the heart (doi: 10.3389/fpara.2023.1244604), we showed that at 84 days post infection there was significant reduction in cardiac inflammatory markers, but only Aspirin treatment significantly reduced cardiac fibrosis. However, we have previously shown that in our mouse model, cardiac fibrosis is progressive (doi: 10.1161/JAHA.119.013365). Thus, it is possible that any potential anti-fibrotic effects may not yet be fully evident at this time point. Including additional time points—both immediately after treatment and at later intervals—would provide a more complete understanding of the kinetics of fibrosis and immunomodulation. We plan to address this in future studies through time-course experiments to capture both the short- and long-term impact of AIP treatment. 

This was noted in the manuscript.

The representative gating strategy for the flow cytometry data was absent. A subset of the population exhibits a remarkably low percentage. What is the number of events acquired for each sample to evaluate these populations? How many cells were analyzed per animal?

Response: We appreciate this observation. On average, 182,148 events were acquired per sample (SEM = 11,738). We generated a supplementary figure 1 and it has been noted in the Results section. 

Histopathological analysis was performed in only 4 representative images? This method should be better described: to evaluate this representative portion of the organ (1/4 of the liver), serial sections, 10-30 micrometers apart, should be analyzed in relation to cells count and tissue area of images.

Response: We appreciate the reviewer’s suggestion. The manuscript has been revised to clarify that histopathological analysis was performed on serial sections representing approximately one-quarter of the liver. While this approach provides a representative evaluation of inflammation and fibrosis and a small, supporting dataset for the study, we acknowledge that, given the liver’s large size, evaluating only a portion may not capture the full heterogeneity of the tissue. In future studies, we plan to analyze multiple liver sections from each animal to provide a more comprehensive assessment. 

The liver lysate was obtained with RIPA Buffer; however, the authors have to mention if they used phosphatase and protease inhibitors, which are essential for preserving the integrity of proteins for subsequent analysis, particularly for the detection of phosphorylated proteins.

Response: RIPA buffer was used to maintain consistency with the cardiac evaluation of these proteins, and phosphatase and protease inhibitors were not included in this study. However, all the liver sections were snap-frozen immediately upon collection and used promptly for protein isolation, which likely prevented detectable protein degradation and did not interfere with the assays performed. We appreciate the reviewer’s suggestion and will incorporate phosphatase and protease inhibitors in future studies to enhance sensitivity and optimize results. 

This has been clarified to indicate that tissues were snap-frozen and processed immediately for the assays in the Methods section for liver collection.

The parasite burden was evaluated in the liver. Do the authors have the information of blood parasite count of this T. cruzi strain in this mouse model along time? This information could facilitate a more comprehensive understanding of the course and stage of infection.

Furthermore, the systemic parasitemia could facilitate the comprehension of the immunological response manifested in the spleen and serve as a correlate with the hepatic parasite burden identified.

Response: We thank the reviewer for this suggestion. Since this study is a follow-up to our recently published cardiac evaluation, parasitemia data as well as cardiac parasite burdens for this T. cruzi strain in this mouse model are described in the previous paper (doi: 10.3389/fpara.2023.1244604). We found that parasitemia levels and cardiac parasite burdens were not significantly different between treatment groups. Thus, we concluded that the immunologic differences between AIP-1/AIP-2 treated groups, and infected untreated controls were due to the direct immunomodulatory effects of these proteins rather than indirect effects that alter parasite levels. 

This is noted in the Results section - “Parasite Load, Hepatic Cellular Infiltration, and Fibrotic Changes”.

Similarly, the researchers had the option of measuring blood cytokines to create a correlation with the systemic data obtained from the spleen.

Response: As part of our cardiac evaluation, we assessed systemic cytokine levels in the serum and found that infected untreated mice exhibited elevated IFN-γ and IL-10 and reduced IL-2. Treatment with AIP-1 restored IL-2 levels without affecting the other cytokines. We also measured serum IgG levels specific for AIP-1 and AIP-2. This data is presented in Figure 5 of our previous publication (doi: 10.3389/fpara.2023.1244604). 

This is noted in the manuscript.

The authors conducted an evaluation of the total and activated STAT-3 protein, which is traditionally activated by IL-6. However, the present model demonstrated a decrease in IL-6 levels in response to the infection.

Response: We thank the reviewer for this comment. While STAT-3 is classically activated by IL-6, it can also be triggered by other cytokines such as IL-10 (doi: 10.3389/fimmu.2023.1160719). In our previous cardiac study, we did see elevated levels of IL-10 in the serum with infection, which could have contributed to increased pSTAT3 levels in the liver, despite liver tissue levels of IL-10 being reduced by T. cruzi infection. Since IL-6 and IL-10 can be produced by both immune and non-immune cells within the liver, future studies will identify the composition of immune cells within the liver to better define the cytokine producing cells and the impact of AIP-1 and AIP-2 on immune cells and non-immune cells within the liver. 

This is noted in the Discussion section.

Moreover, IFN-g exhibited diminished levels during infection, yet these levels were restored following treatment with both AIP-1 and AIP-2. The researchers didn’t evaluate STAT-1 protein, which is known to be activated in response to IFN-g. It would be interesting whether this approach will be further explored in subsequent studies. The STAT-1 gene expression does not necessarily reflect protein levels or activation.

Response: We thank the reviewer for this comment. While IFN-γ levels were restored by AIP-1 and AIP-2, STAT-1 protein and activation were not assessed in this study. We acknowledge that gene expression alone may not reflect STAT-1 activity, and future studies should evaluate STAT-1 and phosphorylated STAT-1 to better understand IFN-γ–mediated signaling in response to these treatments.

This limitation has now been noted in the manuscript.

The Ref [24] did not describe GAPDH primers. Instead of that reference, I suggest the following https://doi.org/10.1038/emi.2013.50.

Response: We thank the reviewer for the suggestion; the reference has been updated as suggested.

## Closing

In addition to this response letter, we are submitting the revised manuscript and supplemental data file with the changes highlighted for review. We feel that the revisions to the manuscript based on reviewer critique have greatly improved the manuscript. We thank you for your continued review and welcome any additional comments or critiques as you consider this manuscript for publication.

Sincerely,

Maria Jose Villar, MSc

National School of Tropical Medicine

Baylor College of Medicine

Texas Children’s Hospital Center for Vaccine Development

Kathryn M. Jones, DVM, PhD

National School of Tropical Medicine

Baylor College of Medicine

Texas Children’s Hospital Center for Vaccine Development

---

## [Reviewer Report · REVIEWERS COMMENTS]

## REVIEWER #1

The manuscript is very interesting and is ready to be published. The abstract, methodology, results, discussion and references are adequate. These results will help further research on this field.